Weekend effect in upper gastrointestinal bleeding: a systematic review and meta-analysis

Shih Pei-Ching 1
Liu Shu-Jung 2
Li Sung-Tse 3 4
Chiu Ai-Chen 1
Wang Po-Chuan 5
Liu Lawrence Yu-Min drlawrenceliu@gmail.com 6 7
1 Department of Family Medicine, Hsinchu MacKay Memorial Hospital , Hsinchu City , Taiwan
2 Medical Library, Tamshui MacKay Memorial Hospital , New Taipei City , Taiwan
3 Department of Pediatrics, Hsinchu MacKay Memorial Hospital , Hsinchu City , Taiwan
4 Graduate Institute of Business Administration, College of Management, Fu Jen Catholic University , New Taipei City , Taiwan
5 Division of Gastroenterology, Department of Internal Medicine, Hsinchu MacKay Memorial Hospital , Hsinchu City , Taiwan
6 Division of Cardiology, Department of Internal Medicine, Hsinchu MacKay Memorial Hospital , Hsinchu City , Taiwan
7 Department of Medical Science & Institute of Bioinformatics and Structural Biology, National Tsing Hua University , Hsinchu City , Taiwan
Lee Yeong Yeh
Electronic publication date: 2018 Jan 12
Publication date: 2018
Volume: 6
Electronic Location ID: e4248
Received 2017 Sep 11; Accepted 2017 Dec 16
Copyright: ©2018 Shih et al.
Copyright year: 2018
Copyright holder: Shih et al.
License: This is an open access article distributed under the terms of the Creative Commons Attribution License, which permits unrestricted use, distribution, reproduction and adaptation in any medium and for any purpose provided that it is properly attributed. For attribution, the original author(s), title, publication source (PeerJ) and either DOI or URL of the article must be cited.
License URL: https://creativecommons.org/licenses/by/4.0/

Keywords: Weekend effect, Systematic review, Upper gastrointestinal bleeding, Meta-analysis, Mortality, Endoscopic intervention

Funding: The authors received no funding for this work.

==============================
Aim

To perform a systematic review and meta-analysis of the weekend effect on the mortality of patients with upper gastrointestinal bleeding(UGIB).

Methods

The review protocol has been registered in the PROSPERO International Prospective Register of Systematic Reviews (registration number: CRD42017073313) and was written according to the Preferred Reporting Items for Systematic Reviews and Meta-Analyses (PRISMA) statement. We conducted a search of the PUBMED, COCHRANE, EMBASE and CINAHL databases from inception to August 2017. All observational studies comparing mortality between UGIB patients with weekend versus weekday admissions were included. Articles that were published only in abstract form or not published in a peer-reviewed journal were excluded. The quality of articles was assessed using the Newcastle-Ottawa Scale. We pooled results from the articles using random-effect models. Heterogeneity was evaluated by the chi-square-based Q-test and I2test. To address heterogeneity, we performed sensitivity and subgroup analyses. Potential publication bias was assessed via funnel plot.

Results

Eighteen observational cohort studies involving 1,232,083 study patients were included. Weekend admission was associated with significantly higher 30-day or in-hospital mortality in all studies (OR = 1.12, 95% CI [1.07–1.17], P < 0.00001). Increased in-hospital mortality was also associated with weekend admission (OR = 1.12, 95% CI [1.08–1.17], P < 0.00001). No significant difference in in-hospital mortality was observed between patients admitted with variceal bleeding during the weekend or on weekdays (OR = 0.99, 95% CI [0.91–1.08], P = 0.82); however, weekend admission was associated with a 15% increase in in-hospital mortality for patients with non-variceal bleeding (OR = 1.15, 95% CI [1.09–1.21], P < 0.00001). The time to endoscopy for weekday admission was significantly less than that obtained for weekend admission (MD = −2.50, 95% CI [−4.08–−0.92], P = 0.002).

Conclusions

The weekend effect is associated with increased mortality of UGIB patients, particularly in non-variceal bleeding. The timing of endoscopic intervention might be a factor that influences mortality of UGIB patients.

Introduction

Upper gastrointestinal bleeding (UGIB) is a significant medical condition worldwide, with an incidence that varies from 48 to 160 cases per 100,000 individuals per year (Tielleman, Bujanda & Cryer, 2015). UGIB is also associated with high morbidity and mortality (Tielleman, Bujanda & Cryer, 2015) and is the most common reason for which gastroenterologists are consulted (Khamaysi & Gralnek, 2013). Advances in endoscopic hemostasis and pharmacotherapy have greatly improved clinical outcomes in these patients (Abougergi et al., 2015). Early endoscopic intervention to diagnose and treat the bleeding sites and close monitoring in the ward are often required, particularly for patients who are considered high risk. Many predictive risk scores for UGIB are available and are primarily based on patient characteristics and endoscopic findings (Das & Wong, 2004).

The weekend effect, defined as increased mortality in patients who are admitted on the weekend (Bell & Redelmeier, 2001; Cram et al., 2004), is suggested to play an important role in patients with UGIB for several reasons, including reduced staff availability, lack of gastrointestinal subspecialists to conduct timely diagnostic or therapeutic procedures, inadequate intensive care facilities (Zhou et al., 2016) and the fact that sicker patients tend to be admitted on the weekend (Shaheen, Kaplan & Myers, 2009). However, the results reported in the literature regarding the association between weekend admission and mortality in patients with UGIB are conflicting (Dorn et al., 2010; Abougergi, Travis & Saltzman, 2014; Tufegdzic et al., 2014; Ahmed et al., 2015). A previous meta-analysis explored the weekend effect of GIB but did not further differentiate different etiologies of UGIB or the timing of endoscopic intervention (Zhou et al., 2016). Therefore, we performed a comprehensive systematic review and meta-analysis of existing cohort studies to evaluate the overall weekend effect on the mortality of patients with UGIB. Understanding the factors that might contribute to increased mortality in patients with UGIB who are admitted over the weekend might aid the development of risk stratifications and the devising of better patient management strategies to improve clinical outcomes.

Materials and Methods

The review protocol has been registered in the PROSPERO International Prospective Register of Systematic Reviews (registration number: CRD42017073313) and was written according to the Preferred Reporting Items for Systematic Reviews and Meta-Analyses (PRISMA) statement (Moher et al., 2010) (Table S1).

Search strategy

Two reviewers (PCS and SJL) conducted a comprehensive search of the databases PUBMED, COCHRANE, EMBASE via OVID, and CINAHL via EBSCO from inception to August 2017, with no language restrictions. The search strategy is provided in Table S2. Controlled vocabularies supplemented with free keywords were used to search for articles on the weekend effect in patients with UGIB. Furthermore, we manually searched relevant references within articles identified in the screening process.

Study selection

Two reviewers (PCS and LYL) independently screened the search results, and disagreements were resolved by discussion with a third reviewer (PCW). We included all observational studies (prospective cohort or retrospective cohort studies) that compared mortality between UGIB patients with weekend versus weekday admission as indicated by the title or abstract. Articles that were published only in abstract form or not published in a peer-reviewed journal were excluded after a full-text assessment of the article.

Data extraction and quality assessment

Two reviewers (PCS and STL) extracted and collected the data independently. Disagreements or uncertainties were resolved after discussion. We recorded the data as follows: first author, publication year of the article, country, study period, study type, enrolled patient type, number of enrolled patients, definition of weekend, and outcomes. We extracted mortality by identifying the number of deaths in each group, and the time to endoscopy was extracted based on the mean with standard deviation or median with interquartile range. Article quality was assessed using the Newcastle-Ottawa Scale (NOS), which consists of three domains (four points for selection, two points for comparability, and three points for outcome). The maximum NOS score for a cohort study was nine points. Studies with more than seven points were considered to be of high quality.

Statistical analysis

We conducted a meta-analysis of the included articles using Review Manager, version 5.3. We used the random-effect model to pool results from articles, accounting for variance among studies. The association between weekend admission and mortality for UGIB patients was assessed by a pooled odds ratio (OR) with 95% confidence intervals (CI). The association between weekend admission and time to endoscopy was assessed by a pooled mean difference (MD) with 95% CI, which was estimated using the mean and standard deviation (SD). If an article reported a median with interquartile range (IQR), we estimated the mean and SD using the median and the formula SD = IQR/1.35 as recommended in the Cochrane Handbook for Systematic Reviews of Interventions (Higgins & Green, 2011). The significance of the pooled OR and MD was determined by a Z test, and we considered P values <0.05 to be statistically significant. We evaluated heterogeneity using the chi-square-based Q-test and I2 test (Higgins & Thompson, 2002). A PQ value <0.05 and I2 >50% indicated the possibility of significant heterogeneity (Higgins et al., 2003). To address heterogeneity, we performed sensitivity and subgroup analyses. In the sensitivity analysis, we omitted one study at a time to evaluate the robustness of the results. A subgroup analysis was conducted according to the study design. We assessed potential publication bias via a funnel plot.

Results

Search results

We retrieved 669 relevant articles through our search strategy and fully reviewed 39 articles. Eighteen articles were included in our systematic review, and 17 articles were included in the meta-analysis (one of the included articles had no available mortality outcome Ahmed et al., 2015) (Fig. 1).

Figure 1 Process for selecting eligible studies.

Table 1 Characteristics of the selected studies.

First author, year	Country	Study period	Study type	Patient type	Total No. of patients	Weekend definition	NOS score	
Weeda, 2016	United States	2010/1–2012/12	Retrospective cohort	UGIB	119,353	Friday midnight to Sunday midnight	9	
Ahmed, 2015	United Kingdom	2000/1–2009/10	Retrospective cohort	UGIB	73,834	Saturday to Sunday	7	
Al-Qahatani, 2015	Saudi Arabia	2005/1–2013/7	Retrospective cohort	Variceal bleeding	937	Wednesday 16:00 to Friday midnight	9	
Wu, 2014	Taiwan	2009/1–2011/3	Retrospective cohort	Peptic ulcer bleeding	744	National holidays and Saturday to Sunday	7	
Tufegdzic, 2014	Serbia	2002/1–2012/1	Retrospective cohort	UGIB	493	Friday 15:00 to Monday morning 7:00	8	
Abougergi, 2014	United States	2009	Retrospective cohort	UGIB	202,259	Saturday 12:00 to Sunday 23:59	9	
Youn, 2012	South Korea	2007/1–2009/12	Retrospective cohort	Peptic ulcer bleeding	388	Friday midnight to Sunday mid-night	8	
Byun, 2012	South Korea	2005/1–2009/2	Retrospective cohort	Variceal bleeding	294	Friday midnight to Sunday midnight	8	
Tsoi, 2012	Hong Kong	1993–2005	Prospective cohort	Peptic ulcer bleeding	8,222	Sunday and public holidays	8	
Haas, 2012	United States	2008/1–2008/10	Retrospective cohort	UGIB	174	Friday 17:00 to Sunday midnight	7	
Groot, 2012	Netherlands	2009/10–2011/9	Prospective cohort	UGIB	571	Friday 23:00 to Monday 7:59 and official holidays	8	
Button, 2011	United Kingdom	1999–2007	Prospective cohort	UGIB	24,421	Saturday to Sunday and public holidays	9	
Jairath, 2011	United Kingdom	2007/5–2007/6	Prospective cohort	UGIB	6,749	Friday midnight to Sunday midnight	9	
Dorn, 2010	United States	1998–2003	Retrospective cohort	UGIB	98,975	Saturday 12:01 to Sunday 23:59	9	
Ananthakrishnan, 2009	United States	2004	Retrospective cohort	UGIB	419,939	Friday midnight to Sunday midnight	9	
Myers, 2009	United States	1998–2005	Retrospective cohort	Variceal bleeding	36,734	Saturday to Sunday	9	
Shaheen, 2009	United States	1993–2005	Retrospective cohort	Peptic ulcer bleeding	237,412	Saturday to Sunday	9	
Schmulewitz, 2005	United Kingdom	2001	Retrospective cohort	UGIB	584	Friday midnight to Sunday midnight and holidays	8	

Study characteristics

Detailed characteristics of the selected studies are listed in Table 1. Seven studies were performed in the United States, six studies were conducted in Europe (United Kingdom, Serbia, and the Netherlands), and the remaining five studies were conducted in Asia (Saudi Arabia, Taiwan, South Korea, and Hong Kong). Of the 18 observational cohort studies, four were prospective, and the others were retrospective. Eleven studies enrolled patients with UGIB, three studies enrolled only patients with variceal bleeding, and the remaining four studies enrolled only patients with non-variceal bleeding. The total number of patients was 1,232,083. Four studies enrolled over 100,000 patients, whereas five studies enrolled between 1,000 and 100,000 patients, and nine studies enrolled <1,000 patients. In 12 studies, the definition of the weekend was Saturday to Sunday; in the other five studies, the weekend was defined as Saturday to Sunday and holidays. One study defined the weekend as Wednesday afternoon to Friday midnight.

Mortality

All studies assessed mortality as an outcome, although in one study, the outcome information was not available. Three of the studies recorded 30-day mortality, and the remaining 14 studies recorded in-hospital mortality. The total number of pooled study patients was 1,158,249. Weekend admission was associated with significantly higher 30-day or in-hospital mortality in all studies (OR = 1.12, 95% CI [1.07–1.17], P < 0.00001) (Fig. 2). Increased in-hospital mortality was also associated with weekend admission (OR = 1.12, 95% CI [1.08–1.17], P < 0.00001) (Fig. 3). Due to the observed heterogeneity between the studies, we conducted a subgroup analysis. To identify the source of heterogeneity, we constructed our analysis using two main categories based on the etiology of UGIB. No significant difference in in-hospital mortality was observed between patients admitted with variceal bleeding during the weekend versus weekdays (OR = 0.99, 95% CI [0.91–1.08], P = 0.82) (Fig. 4). In contrast, weekend admission was associated with a 15% increase in in-hospital mortality in patients with non-variceal bleeding (OR = 1.15, 95% CI [1.09–1.21], P < 0.00001) (Fig. 5). We also omitted one study at a time to evaluate the robustness of the results, and the pooled results still showed no difference.

Figure 2 Forest plot of odds ratio for 30 day or in-hospital mortality due to UGIB during weekend versus weekday.

Figure 3 Forest plot of odds ratio for in-hospital mortality due to UGIB during weekend versus weekday.

Figure 4 Forest plot of odds ratio for in-hospital mortality due to variceal bleeding during weekend versus weekday.

Figure 5 Forest plot of odds ratio for in-hospital mortality due to non-variceal bleeding during weekday versus weekend.

Time to endoscopy

Eight studies assessed differences in time to endoscopy. The time to endoscopy for weekday admission was significantly less than that for weekend admission (MD = −2.50, 95% CI [−4.08–−0.92], P = 0.002) (Fig. 6). However, a high degree of heterogeneity was found in this pooled result. Because only eight studies were included, it was difficult to conduct further subgroup analyses based on the study design.

Figure 6 Forest plot of mean difference for time to endoscopy during weekday versus weekend.

Publication bias

Visually apparent asymmetry was present in the funnel plot of mortality that included all studies (Fig. 7), which might indicate that small studies produced bias in the random-effect model. However, we pooled mortality data using a fixed-effect model, which showed no significant difference in ORs between the two models.

Figure 7 Funnel plot assessed bias of selected studies.

Discussion

Our study constitutes the first systematic review and comprehensive meta-analysis to address whether a global weekend effect is associated with mortality in patients with UGIB. We found a significant weekend effect on mortality, a finding that is consistent with several nationwide studies in the United States and population-based studies in the United Kingdom (Dorn et al., 2010; Button et al., 2011; De Groot et al., 2012; Ahmed et al., 2015; Weeda et al., 2016). We also observed the same result after we pooled the data from another nationwide study (Ananthakrishnan, McGinley & Saeian, 2009). Several factors associated with healthcare resources have been suggested as possible reasons for increased mortality on weekends. On the weekend, fewer staff are available to carry the same workload compared with weekdays (Tarnow-Mordi et al., 2000), healthcare providers are less experienced (Meltzer et al., 2002), specialists such as gastrointestinal subspecialists and interventional radiologists are limited (Hearnshaw et al., 2010), and continuity of care is poor (Petersen et al., 1994); these factors are all related to a higher risk of adverse outcomes. Another possibility is that patients admitted on the weekend might have more severe illness with increased co-morbidities (Mikulich et al., 2011). It is also conceivable that patients with minor symptoms on the weekend are tempted to wait and visit their family physicians on a weekday; one study in the United States reported that patients are admitted on the weekend at a lower rate than on weekdays (Dorn et al., 2010). However, the relative contributions of these factors remain unclear. A cross-sectional analysis did not find a correlation between the availability of specialists and the increased mortality risk observed with a weekend admission (Aldridge et al., 2016). One multicenter prospective study attempted to explore the “off-hours” staffing issue and found that a more important factor influencing mortality was disease severity (De Groot et al., 2012). Nevertheless, another study found that a higher Charlson comorbidity score was not associated with a significant weekend effect after adjusting for co-morbidity (Ahmed et al., 2015).

To explain the moderate to high heterogeneity detected in our results, we conducted a subgroup analysis of 30-day and in-hospital mortality and different etiologies of UGIB. We discovered that the weekend effect was only significantly associated with in-hospital mortality but did not reach significance for 30-day mortality. This result might be explained in part by the possibility that 30-day mortality is influenced not only by UGIB but also other organ diseases (Tsoi et al., 2012). We also found that the weekend effect was primarily associated with non-variceal bleeding. In the two largest studies of patients with non-variceal bleeding from the United States, a significant weekend effect was found (Ananthakrishnan, McGinley & Saeian, 2009; Shaheen, Kaplan & Myers, 2009). Studies of patients admitted due to variceal bleeding in the United States, United Kingdom and Asia all showed no difference in mortality between weekend and weekday admission (Ananthakrishnan, McGinley & Saeian, 2009; Myers, Kaplan & Shaheen, 2009; Nahon et al., 2010; Byun et al., 2012; Al-Qahatani et al., 2015). Because patients with variceal bleeding often had more risk factors, such as more severe liver disease estimated by the model for end-stage liver disease (MELD) score (Bambha et al., 2008), portal hypertension, hepatocellular carcinoma, and circulatory dysfunction (Garcia-Tsao, Bosch & Groszmann, 2008), the time of admission played only a small role in mortality.

As previously observed, the limited availability of specialists to perform timely endoscopic diagnosis and intervention is another important issue for UGIB patients (Wysocki, Srivastav & Winstead, 2012) and might also be associated with the weekend effect. In real-world practice, it is likely that fewer gastrointestinal subspecialists and other staff required for endoscopic procedures are on-call on weekends (Bell & Redelmeier, 2001). We assessed the difference in time to endoscopy between weekend and weekday admission. In our study, an analysis of pooled data showed that time to endoscopy during weekday admission was significantly less than that for weekend admission, indicating that endoscopy timing might be a critical factor. This finding was present in the majority of studies, which indicated significant delays to endoscopy on the weekend (Schmulewitz, Proudfoot & Bell, 2005; Ananthakrishnan, McGinley & Saeian, 2009; Myers, Kaplan & Shaheen, 2009; Shaheen, Kaplan & Myers, 2009; Dorn et al., 2010; Button et al., 2011; Jairath et al., 2011; Youn et al., 2012; Abougergi, Travis & Saltzman, 2014; Al-Qahatani et al., 2015). However, more recent guidelines recommending that endoscopy should be performed for both variceal and non-variceal bleeding might have steered healthcare system policy toward early endoscopy for the management of UGIB (Barkun et al., 2010; Quraishi, Khan & Tripathi, 2016). As a result, the impact of time to endoscopy on the mortality of patients admitted on the weekend might have lessened (De Groot et al., 2012; Haas, Gundrum & Rathgaber, 2012; Tsoi et al., 2012; Tufegdzic et al., 2014; Wu et al., 2014).

Our meta-analysis showed increased mortality in patients with UGIB and delayed time to endoscopy during the weekend; however, these results still exhibit moderate to high heterogeneity even after subgroup analyses. Although a previous study showed that 45% of patients with variceal bleeding underwent endoscopy on the day of admission versus only 33% of those with peptic ulcer disease (Shaheen, Kaplan & Myers, 2009), whether time to endoscopy explains increased in-hospital mortality in non-variceal bleeding remains to be determined. Thus, there are conflicting results regarding the weekend effect on UGIB mortality and time to endoscopy, and several possible explanations could explain this phenomenon. First, the guidelines for managing UGIB are inconsistent between regions and even between individual hospitals within the same country. For example, a large mean difference in time to endoscopy was found in the United Kingdom because only 52% of hospitals had a formal out-of-hours endoscopy team (Jairath et al., 2011). Second, we could not distinguish the initial severity of disease because studies had extracted data from national databases with limited clinical details of patient characteristics. Third, in the majority of studies, in-hospital mortality rather than 30-day mortality was the endpoint. The length of follow-up required to evaluate the weekend effect on mortality is unclear. Whether these variables significantly influence studies of the weekend effect must be considered, and a more comprehensive study is needed in the future.

Some limitations of this systematic review should be noted. The results in our meta-analysis exhibited moderate to high heterogeneity, which might reduce their validity. Potential patient overlap may also have occurred among different cohorts, although we do not have a quantitative method to determine the extent of this overlap. Because all data were extracted from observational studies, non-randomized patient characteristics might confound the outcome of mortality between weekend and weekday admission. Although previous studies have attempted to determine a difference in the severity of illness for weekday and weekend admissions, no accurate tool exists for measuring disease severity at the time of admission. The increasing use of electronic national early warning scores (Royal College of Physicians, 2015) might solve this problem. Furthermore, focusing on the weekend effect on mortality as the primary outcome might fail to reflect the quality of medical care because only 4% of deaths are thought to be avoidable (Hogan et al., 2015). Some studies discussed other outcome measures such as the re-bleeding rate, length of hospital stay and hospital charges; however, we did not extract these outcomes because too few studies provided sufficient data to conduct a meta-analysis. Similarly, we were also unable to determine whether there is a difference between variceal and non-variceal bleeding in time to endoscopy due to limited data. The above confounding factors and limitations might influence the pooled results and require a careful interpretation of our meta-analysis results.

Conclusions

A significant weekend effect is associated with mortality in patients with UGIB worldwide, particularly in cases of non-variceal bleeding. The time to endoscopy for weekday admissions is less than that for weekend admissions. This finding suggests that timing to endoscopic examination and appropriate intervention might be important factors that influence mortality in UGIB patients.

Supplemental Information

Table S1 PRISMA checklist

Click here for additional data file.

Table S2 Search strategy

Click here for additional data file.

Table S3 NOS assessment

Click here for additional data file.

Table S4 Raw data

Click here for additional data file.

Article S1 Rationale and contribution of the meta-analysis

1. The rationale for conducting the meta-analysis; 2. The contribution that the meta-analysis makes to knowledge in light of previously published related reports, including other meta-analyses and systematic reviews.

Click here for additional data file.

This manuscript was edited for English by American Journal Experts (AJE).

Additional Information and Declarations

Competing Interests

Author Contributions

Data Availability

The authors declare there are no competing interests.

Pei-Ching Shih performed the experiments, analyzed the data, wrote the paper, prepared figures and/or tables.

Shu-Jung Liu performed the experiments, contributed reagents/materials/analysis tools, wrote the paper.

Sung-Tse Li analyzed the data, contributed reagents/materials/analysis tools, prepared figures and/or tables.

Ai-Chen Chiu analyzed the data, reviewed drafts of the paper.

Po-Chuan Wang conceived and designed the experiments, contributed reagents/materials/analysis tools, prepared figures and/or tables, reviewed drafts of the paper.

Lawrence Yu-Min Liu conceived and designed the experiments, reviewed drafts of the paper.

The following information was supplied regarding data availability:

The raw data is provided as a Supplemental File.

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
