# Peer review of "Weekend effect in upper gastrointestinal bleeding: a systematic review and meta-analysis"

_PeerJ, doi:10.7717/peerj.4248_

## Round 0.1 · original submission · Major Revisions

Overall, the comments from reviewers were minor. However, there remains a concern on language despite editing, and I strongly recommend authors to look again at their manuscript in more detail since I have noticed a few spelling mistakes myself.

In addition, for time to endoscopy, it is unclear in the text if there is a difference between variceal and non-variceal bleeding, and this is important especially the results showed a difference in mortality between the two. I also agree with other reviewers that reporting of studies is extremely important, and this should be given in more details and explained if necessary (for example Jairath 2011, why is there a large mean difference in time to endoscopy?).

·

Basic reporting

Report written using a good standard of English. Concise and clear use of language to convey the intended message effectively. Use of professional help for manuscript editing noted and commendable.
Seven figures and one table included in the article.
Figure 1 shows flowchart of selecting eligible studies.
Figure 2 shows Forest plot of OR for total mortality (30-day and in-hospital mortality) due to UGIB during weekend versus weekday.
Figure 3 shows Forest plot of OR for in-hospital mortality only.
Figure 4 shows Forest plot of OR for in-hospital mortality in a subset of patients with UGIB due to variceal bleed.
Figure 5 shows Forest plot of OR for in-hospital mortality in a subset of patients with UGIB due to non-variceal bleed.
Figure 6 shows Forest plot of mean time to endoscopy during weekday versus weekend.
Figure 7 shows Funnel plot assessing bias in study.
Table 1 summarises the pertinent points from the selected studies.
All included figures and tables relevant to the reporting of the study and included appropriately.
Of note, there is a mention of another table labelled S2 which is supposed to list the search strategy, which has been omitted from the reviewers copy of the article.

Experimental design

Study was well designed and follows the expected standards for systematic review and meta-analyses.
Study aims to assess the effect of weekend admissions on UGIB and the resulting outcome of mortality, which is a poorly understood phenomenon and is not widely reported.
Study question framed clearly and subsequent study reporting clearly focuses on addressing the study question.

Methodology reported well, in a clearly reproducible manner. However, ‘Table S2’ which supposedly lists the search strategy was not included in the manuscript for reviewer.

The standard steps for performing systematic review was followed e.g. searching and screening by 2 different reviewers separately and the involvement of a third reviewer to resolve any disagreement.
The major databases were comprehensively searched for without any language restrictions. Articles published in abstract only form or in non-peer review journals were excluded. There was also no mention of searches performed in the ‘grey literature’. However, although limiting the search to only peer reviewed journals from the major databases may restrict the number of studies that can be included, it may partly ensure that only higher quality studies with complete data are included.

Validity of the findings

Data reported in a clear and relevant manner with selected Forest plots of interest included.
Study findings were discussed in a fair manner, considering the effect of heterogeneity where relevant.
The conclusion is well stated and addresses the study question.
Given the small number of studies meeting the inclusion criteria for this systematic review and the specific issues highlighted by the authors e.g. use of non-standardized definition for weekend, use of different primary end-points and reporting of time to endoscopy only in some studies; this study raises more questions than answers.
More robust well-designed prospective studies need to conducted in multiple centres worldwide to examine the weekend effect on UGIB admission to provide more conclusive findings.
Overall the effort of the authors in conducting and reporting this study is commendable.

Additional comments

Sound methodology and clear reporting of results.
Good discussion with explanations for observed results backed by current evidence.
Meets the required standard for publishing.

·

Basic reporting

Comment 1. Please check grammatical error. For example, line 207 ” admission showed significantly shorted times”

Experimental design

no comment

Validity of the findings

Comment 2. There were only seven studies shown in figure 5 for the increased in-hospital mortality due to non-variceal bleeding in weekend. And with eight studies recruited in figure 6 showing time to endoscopy during weekday is shorter. Yet, only two studies overlapped in these two figures (Wu, 2014 and Youn, 2012). So, whether longer time to endoscopy (figure 6) may explain the increased in-hospital mortality due to non-variceal bleeding (figure 5) may be a question. Because the initial settings of patients were different. Ideally, It would be better if all the recruited studies are the same in figure 5 and figure 6.

Additional comments

Comment 3. Line 227 to line 228, I don’t think the different definition of weekend from recruited studies will affect the result. As long as there is a mention that lacking of manpower during the day which was defined as weekend. Are there any explanation of the definition of their weekend (for instance, less manpower in the subspecialty during the defined weekend) in the quoted articles?

Reviewer 3 ·

Basic reporting

I would like to applaud the author for registering their protocol PROSPERO and following the PRISMA guideline. Many systematic reviews suffer from poor reporting and inadequate search methodologies. I was pleasantly surprised to see this systematic review was well reported and there was no major concern regarding the search strategies.

a few minor suggestions:
1) When reporting the databases, it is important to report the interface and the coverage for each database [line 25 and 26]. Searching the same database via different interfaces can lead to different results. For example, EMBSE is available via OVID or embase.com. CINHAL is available via EBSCO, OVID, and ProQuest.

2) It is recommended to report the number of articles you retrieved per line of search query.

3) It is also a best practice to report any software you use for screening and/or citation management

Experimental design

1) In PubMed search, Line#6 and Line #17, there are typos: admission tim to admission time. I hope these were mistakes in reporting and not the actual search strategy.

2) In PubMed search, remove extra line (line #16) from the search. Otherwise, it might be misleading or confusing to the readers.

A general comment: Databases are very structured and are meant to be searched by controlled vocabularies or index term. It might be important to use key mesh terms in your search. For example, Time-to-treatment or Time Factors could be very good mesh terms to add to your search. To increase the search sensitivity.

Validity of the findings

No comment

Additional comments

No comment

---

## Round 0.2 · accepted · Accept

Thank you for making efforts to improve the language of manuscript. The comments from reviewers have been addressed adequately and I am happy to accept the paper in its current revised form.